# Learning and Interpreting Multiple Representations of Semantics in a Neurobiological System

## Abstract

A defining feature of computation in the human brain is that different regions can manifest different representations of the same object set. Here we introduce a novel method to learn and interpret multiple neural representations of lexical objects within specific, topographically-defined brain areas. Our approach fine-tunes a pre-trained language model (LM) for each brain region of interest, resulting in better alignment of the LM's representational space with that of the corresponding brain area. This alignment is achieved through supervised structural pruning of LM features, which selects a subset of features most relevant to the target brain region. We then interpret these retained features using a linear probing task to identify the semantic information they encode. Both the pruning and probing steps are validated through out-of-sample testing, with pruning significantly improving the prediction of brain representations. This method advances on existing approaches by *i*) eliminating the reliance on hand-crafted encoders, reducing potential biases; *ii*) optimizing the alignment process via data-driven learning; and *iii*) providing interpretability of the semantic features in a black-box LM. From a neurobiological perspective, we find that brain regions encoding social and cognitive aspects of lexical items consistently also represent their sensory-motor features, though the reverse does not hold.

## 1 Introduction

A representational space can be described using basis vectors, which define the dimensions along which objects are positioned based on their values and variances. While artificial neural networks (ANNs) learn representations that are by definition distributed, recent work suggests that they can also develop signatures of modular representational spaces, where different populations of units may encode distinct concepts, categories, or syntactic properties. This seen when certain groups of units show variance that is informative for specific categories, but uninformative for others. For example, in language models it is possible to identify subsets of units that specifically encode syntactic properties or capture semantic dimensions of different categories (e.g., Cao et al., 2021; Manrique et al., 2023), and modular structure has been identified in several studies (e.g., Lepori et al., 2023; Purushwalkam et al., 2019).

These findings suggest but do not prove the converse – that a group of objects could be represented differently across functionally defined modules in ANNs. The main challenge is that representation in ANNs is inherently distributed, making it difficult to isolate functionally specific modules encoding unique representational spaces. Low-rank factorization methods, which extract latent dimensions from the entire object-by-unit activation matrix (Cheng et al., 2017), can obscure smaller, localized modules with limited contributions to overall model covariance. While subspace clustering methods aim to address this (Parsons et al., 2004), they require several heuristics and a-priori decisions (Elhamifar & Vidal, 2013). And of course, without any principled analysis of covariance to guide selection, multi-unit activity cannot be meaningfully interpreted. To illustrate, even a random sampling of ANN units, without using any feature-selection criterion, can produce the appearance that different populations encode different input dimensions. However, this lacks value unless the units systematically encode meaningful semantics. For example, some units may respond

Figure 1: Workflow Overview: The workflow begins with supervised pruning, which selects a subset of language model (LM) features that improve the alignment between word-to-word distances in the LM's representational space and brain representations. In the second stage, a supervised probing task is applied to determine the specific semantic content encoded by the pruned features. The brown and red graphical elements indicate the pruning and probing workflows applied to study representations in two different brain regions.

identically to all objects, or even not respond to any of the objects, providing no useful information. Others units could respond randomly, unrelated to any coherent object representation.

While similar considerations arise when studying representations in biological systems, these may offer clearer insights into representations due to their topographical organization, where spatially adjacent processing units often support similar computations (e.g., vision, audition, language). This spatial structure can aid in identifying proximally-organized neural populations that may encode semantics. Additionally, the spatial resolution of neurobiological recordings often allows modeling single-unit activations using linear encoding models, predicting neural responses based on object features (e.g., Mitchell et al., 2008; Sucholutsky et al., 2023). Taken together, in biological systems, the ability to identify single units and group them based on spatial proximity provides a way to identify neural populations that encode semantic information, and then analyze the representational space inherent in multi-unit activity in this population. Most importantly, we show that when the questions is approached from this perspective, human-interpretable semantic labels can then be assigned to the dimensions underlying these neural representations, in this way identifying the meaning dimensions that organize representations in different brain areas.

As detailed, we introduce a combined method for: 1) identifying meaningful unit populations within the brain, 2) formally capturing the variance dimensions they encode through supervised pruning, and 3) providing interpretable explanations of these variance dimensions via a probing task. Figure 1 presents an overview of the methodology. We begin by identifying brain clusters (functional regions) involved in semantic processing and define the representational space for each cluster. Different brain areas are sensitive to different information dimensions and so will represent lexical items in differ ways. Subsequently, for each brain area, we perform supervised pruning of a language model (GloVe) to align the lexical similarity between words with the representational structure of that brain area. This produces different sets of retained features for different brain regions. In a second step, we use probing to interpret the semantics encoded in these retained features by evaluating how well each of the retained feature sets can predict human annotations for new words. Using this workflow we investigate whether it is possible to identify and interpret multiple, distinct semantic representations for a single set of lexical items, across topographically constrained neural clusters in the human brain. We address three inter-related aims:

**Aim 1** is to evaluate whether it is possible to identify functional brain modules where: (1) the module's representational space — quantified via object-to-object distance matrices derived from multi-unit population activity — can be predicted using object distances from a given language model

$M$; and (2) where said predictions can be optimized, separately for each brain area, by identifying subsets of embedding dimensions within $M$, rather than using the entire model.

**Aim 2**, conditional on the success of Aim 1, the aim is to: (1) determine whether the subspaces identified in Aim 1 are consistent across brain areas, i.e., whether representations in different brain regions are best explained by different latent dimensions in $M$; and (2) provide a human-interpretable explanation of the semantic dimensions that underlie the subspaces in $M$ identified by the optimization algorithm. If successful, this would show that it is possible to interpret in what manner lexical semantics are multiply realized, via different population codes, across the human brain.

**Aim 3**, dependent on the success of Aims 1 and 2, combines the subsets of embedding dimensions learned for different brain areas into a unified graph to determine if communities in the graph code for different semantics.

Our approach addresses two related key limitations in current methods for studying semantic representations in different brain areas. The first improvement is that current approaches require collecting human-annotated feature-ratings for the same stimuli for which neurobiological recording are obtained. These annotations are used to construct interpretable encoding models that align object distances in the model with object distances in different brain areas (e.g., Mitchell et al., 2008; Fernandino et al., 2022). In contrast, our method provides an interpretation of the semantic representations that organize the representational space without requiring human annotations for the items producing the neurobiological responses. As indicated, this is done by using a tailor-pruned version of a black-box model to first approximate object distances in a target brain area, and then interpreting the information encoded in the pruned model through a probing task.

A second improvement is that current methods require manual construction of annotation frameworks, which in turn can be influenced by an experimenter's own subjective decision about which features to include. This biases the analysis towards those few dimensions that an experimenter considers relevant. In contrast, our method is objective in that it uses a generic language model (specifically, a subspace of the model learned via pruning) for encoding a brain region's representational space. In this way we remove the need for manual feature selection or explicit assumptions about the encoding model, taking advantage of the semantics already captured in pre-trained models.

Finally, we show that using this approach it is possible to aggregate information across multiple learned representations by combining the pruned language models (optimized for different brain areas) into a single graph structure. This graph, built directly from the pruned solutions identifies communities of features in the LM that code for different semantics.

## 2 METHODS

### 2.1 DATASETS

The neuro-biological data consisted of functional MRI (fMRI) recordings, which capture brain activity using blood oxygenation level-dependent (BOLD) signals. These recordings provided data for approximately 14,000 virtual sensors (voxels) per participant, sampled at a temporal resolution of 0.5 Hz. Data were collected from nine participants, as reported in Mitchell et al. (2008). The voxel IDs are consistent across participants, meaning that a given voxel ID refers to the same brain region for all participants. This allowed us to combine data across participants. We restricted the analysis to the subset of 13,189 voxels that contain non-null data for all participants. Each of the nine participants was presented with 60 distinct nouns, each repeated six times. For each voxel, this design yields a total of $54$ activation values per noun (9 participants × 6 presentations). The complete data can thus be represented as a four-dimensional tensor $\mathbf{A} \in \mathbb{R}^{9 \times 6 \times 60 \times 13189}$, where the dimensions correspond, respectively to participants, repeated presentations, nouns and voxels. To improve the signal-to-noise ratio, we averaged across the participant and presentation dimensions, collapsing the first two dimensions. This results in a two-dimensional Voxel-Noun matrix $\mathbf{V} \in \mathbb{R}^{13189 \times 60}$, where each element $v_{i,j}$ represents the averaged activation value for voxel $i$ in response to noun $j$. This matrix is referred to as the Voxel-Noun matrix.

Mitchell et al. also provide a simple vector-semantics model, which they show can predict brain activity for the 60 nouns. It represented as a noun-semantics matrix $\mathbf{S} \in \mathbb{R}^{60 \times 25}$, where each of the 60 nouns is encoded as a vector of 25 semantic features. These features were derived from a corpus-

based analysis, where each noun's vector reflects its co-occurrence with 25 manually selected verbs. These 60 nouns were drawn from 12 general categories, with 5 nouns per each. These categories were: body parts, furniture, vehicles, animals, kitchen utensils, types of buildings (e.g., apartment, igloo), parts of buildings, clothing, insects, vegetables, and man made objects. Intuitively, it is possible to represent these 60 nouns on multiple types of semantic dimensions including, for example, visual, auditory, and relation to human-related activities, making them an interesting candidate for the study of representation patterns.

We also used an additional dataset provided by Binder et al. (2016), which provides human ratings for 534 words on 65 semantic dimensions. Participants in that study rated the relevance of each dimension to each word. We used this dataset to probe for information in GloVe embeddings as detailed in Probing section below.

## 2.2 MODEL

As a vector space model, and as a target for Pruning, we used GloVe (Pennington et al., 2014), which provides an adequate choice for predicting human similarity judgments for various concepts (e.g., Chersoni et al., 2021). We extracted GloVe word embeddings for the 60 nouns used by Mitchell et al. These embeddings form a matrix $\mathbf{E}^{(M)} \in \mathbb{R}^{60 \times 300}$, using GloVe with 300 dimensions. Additionally, we extracted GloVe embeddings for 534 words from the Binder et al. dataset Binder et al. (2016), forming a matrix $\mathbf{E}^{(B)} \in \mathbb{R}^{534 \times 300}$. The embeddings for the 60 nouns were used to generate pruned GloVe solutions, and the embeddings for the 534 words were utilized to construct predictive matrices for the Probing analysis.

## 2.3 PRELIMINARIES: IDENTIFYING BRAIN AREAS FOR REPRESENTATION LEARNING

In our study, we use the noun-semantics matrix $\mathbf{S}$ as an encoding model to identify brain regions where a linear mapping from the semantic features to each voxel's activity produced a significant fit. As an initial step, we performed a voxel-wise linear regression analysis, where the activation values for the 60 nouns in each voxel (each row in the Voxel-Noun matrix $\mathbf{V}$) were predicted using the $60 \times 25$ Noun-Semantics matrix $\mathbf{S}$. For each voxel, the encoding model's $R^2$ value was stored. This model was fit separately for each of the 13189 voxels.

We then spatially clustered voxels as follows. The goal of clustering was to identify spatially coherent groups of voxels with good predictive accuracy from the encoding model, which would serve as the focus for subsequent representation learning. As a first step, we selected voxels with regression $R^2$ values in the top-half of a median split. We then created an adjacency matrix, defining two voxels as connected if they shared at least a vertex contact within their 26-neighbor adjacency. Wn then applied agglomerative clustering to this adjacency matrix, with a distance threshold of 10mm between clusters, and selected clusters that contained more than 8 voxels. In summary, this workflow relies on the local topographic organization of brain activity, constraining clustering by spatial contiguity.

This analysis identified a total of 321 activation clusters. To obtain a manageable number of activation clusters, and to account for potential representational similarities among them, we first identified the most representative clusters based on their word-to-word similarity structure. This was performed as follows. For each cluster, we defined a cluster-specific activation matrix $\mathbf{A}_c \in \mathbb{R}^{N_c \times 60}$, where $N_c$ represents the number of voxels in the cluster and 60 corresponds to the words. The word-to-word similarity for each cluster was captured in a cluster-specific similarity matrix $\mathbf{S}_c \in \mathbb{R}^{60 \times 60}$, computed by taking the Pearson correlation between the activation values of all word pairs. We defined prototypical clusters as those whose similarity matrices $\mathbf{S}_c$ were representative of the broader set of clusters. To this end, we first extracted the upper triangle values of all 321 $\mathbf{S}_c$ matrices and arranged them in a 321-row table, which we refer to as the "AllClustersSM" matrix. We then applied agglomerative clustering to AllClustersSM, using a threshold of Pearson correlation $r \geq 0.5$ to group rows (activation clusters) presenting similar representational structures. In other words, this step grouped activation clusters that exhibited similar word-to-word similarity patterns. Next, we retained only cluster groups containing at least three members and identified the prototypical cluster within each group as the one that on average was most strongly correlated with the others in that group. This workflow ultimately produced 22 prototypical activation clusters whose similarity matrices $\mathbf{S}_c$ were representative of the overall brain-wide pattern.

## 2.4 PRUNING METHODS (AIM 1)

As indicated above (section 2.3), prototypical activation clusters were identified using the combined following constraints: linear readout of information of each unit in the module, the units being spatially adjacent, and the representation being prototypical of a set of similar representations independently identified in the brain.

Given an activation cluster's word-to-word similarity matrix $\mathbf{S}_c$, we can assess how well it is predicted by GloVe representations. To do this, we first use the GloVe word embedding matrix $\mathbf{E}^{(M)}$ to generate a GloVe similarity matrix $\mathbf{S}_{\text{GloVe}}$. This matrix is computed by calculating the cosine similarity between the embedding vectors of each word pair in $\mathbf{E}^{(M)}$. We then compute the Spearman correlation (rho) between the upper triangles of $\mathbf{S}_c$ and $\mathbf{S}_{\text{GloVe}}$, which reflects the similarity of the representations produced from GloVe and from the brain data (Kriegeskorte et al., 2008). We consider this value as the baseline representational-alignment between the computational model and brain activity. Note that $\mathbf{S}_{\text{GloVe}}$ is identical in all analyses, whereas $\mathbf{S}_c$ differs for each of the 22 prototypical activation clusters.

To evaluate whether it is possible to learn an improved alignment, we applied a feature pruning algorithm, which selects a subset of GloVe features that produce an improved GloVe-brain representational alignment compared to using the full set of 300 GloVe features. The complete pruning algorithm is presented in Appendix Algorithm 1. The algorithm is based on sequential feature selection, where GloVe features in $\mathbf{E}^{(M)}$ are first ranked by their importance in predicting the brain-derived $\mathbf{S}_c$, and then an optimal subset of features is identified for prediction. This pruning procedure was applied independently to each of the 22 prototypical activation clusters. Specifically, each $\mathbf{S}_c$ matrix supervised a distinct pruning processes, each aiming to discover a subset of GloVe features that outperformed the full feature set.

In summary, rather than using the full $60 \times 300$ matrix, the pruned solutions reflect a reduced feature set of size $d$, where $d < 300$, and the indices of these selected features were stored for downstream analyses. We refer to these subsets as $\mathbf{F}_c$, where $c$ indexes each activation cluster (1 to 22) and $\mathbf{F}_c$ refers to the set of indices for the selected features in cluster.

For each of the 22 clusters we implemented pruning in ways stages. First, as described above, we analyzed the full $60 \times 300$ matrix to identify the optimal number of features $d$, where $d < 300$. Second, we employed a cross-validation (CV) framework, where in each fold, one word was left out, and its corresponding row was removed from both $\mathbf{S}_c$ and $\mathbf{E}^{(M)}$. In this workflow, the CV process was supervised by a $59 \times 59$ similarity matrix derived from brain activity, which was used to prune a $59 \times 300$ GloVe embedding matrix. The learned mapping was then applied to the left-out word, predicting the 59 pairwise similarity values across domains. As a baseline, these 59 similarity values for the left-out word were computed using all 300 features. In this way, CV evaluates whether it is possible to better predict the 59 similarity judgments for the left-out word when using the retained features than when using the full 300-feature set.

## 2.5 PROBING METHODS (AIM 2)

Probing (Belinkov, 2022) evaluated the ability to decode 65 human-annotated semantic feature values from GloVe embeddings. $\mathbf{E}^{(B)} \in \mathbb{R}^{534 \times 300}$ was the GloVe embedding matrix for the 534 words analyzed by Binder et al. (2016), and $\mathbf{Y}^{(\text{Binder})}$ was the human-annotated feature matrix for those same words. Probing quantifies how well the semantic feature values in $\mathbf{Y}^{(\text{Binder})}$ can be predicted from the embeddings in $\mathbf{E}^{(B)}$. The target matrix $\mathbf{Y}^{(\text{Binder})}$ consists of 65 feature annotations for 534 words, capturing semantic dimensions including vision, audition, emotion, and cognition (see Appendix A.1).

We used a Partial Least Squares Regression (PLSR) model to map $\mathbf{E}^{(B)}$ to $\mathbf{Y}^{(\text{Binder})}$. In each cross-validation fold, 533 words were used to train the model, with the test set comprising GloVe embeddings and 65 feature annotations for the left-out word. The learned model was applied to predict the left-out word's 65 features. This process was repeated for all 534 words, generating a $534 \times 65$ prediction matrix, $\mathbf{Z}$, for probing analysis. Probing evaluated the PLSR model by correlating the predicted and ground-truth human-rating values for each of the 65 features. These correlations were

generally high, reproducing Chersoni et al. (2021), see Appendix Figure 4. Note that this performance was obtained when using all 300 GloVe Features.

In the main analysis we applied the learning/testing PLSR procedure to GloVe embeddings constrained to the features from each activation cluster, $\mathbf{F}_c$ (see Figure 1). This allowed us to probe the information encoded in each pruned feature set. Note that pruning was supervised by brain activity data obtained for a different set of words, independent of the probing dataset. With 22 feature sets, this produced 22 vectors of 65 correlation values each.

The analysis is subject to a noise ceiling, because the behavioral ratings forming the prediction target (the $534 \times 65$ matrix, $\mathbf{Y}^{(\text{Binder})}$) are averages over human feature-ratings, which are inherently noisy. The noise ceiling cannot be precisely quantified due to the online data-collection method used in the original study Binder et al. (2016), but is below 1.0. An indication is given by the fact that on average, single-participant ratings and group-ratings showed a median correlation of $R = 0.80$.

## 2.6 Creating a graph from feature-subsets (Aim 3)

We integrated the feature sets selected from the 22 sets $\mathbf{F}_c$ into a single graph. In the graph, nodes were feature indices selected by pruning. Features were connected if they appeared together in the selected feature subsets, and the edge weight between any two features reflected the number of times they co-occurred across subsets. The graph was partitioned using the Louvain algorithm (Blondel et al., 2008), selecting the partition that maximized modularity from 100 runs. This analysis returned four distinct communities, each representing a unique set of GloVe features.

To assess whether the communities found in the partition encode different semantic types, we evaluated each community's features individually. We tested their performance in predicting human similarity judgments on two datasets: Wordsim-353 (Agirre et al., 2009) and Simlex-999 (Hill et al., 2015). We also tested them on a standard analogy benchmark (Mikolov et al., 2013) to determine the semantic content within each community.

# 3 Results

## 3.1 Pruning results

In all 22 activation clusters, predictions of brain representations were always improved using the subset of features learned via pruning. Furthermore, these improvements generalized beyond the training data as shown in cross-validation tests. Table 1 presents the results, for each of the 22 brain areas that constituted a prototypical activation cluster. The table shows the pruning results per cluster when pruning was applied to complete dataset or to out of sample folds in a context of cross-validation.

The majority of brain areas identified were in parieto-temporal areas, lateral temporal and temporal-occipital areas, which are the ones most often implicated in semantic processing in the brain (e.g., Binder et al., 2009). The very few exceptions included the left anterior cingulate cortex, right inferior frontal gyrus, and left postcentral gyrus, but we note the first two produced relatively low correlations.

In nearly all cases, the pruned solutions achieved these improved predictions while retaining fewer than 25% of the GloVe features ($N < 75$ of 300), with several clusters requiring as few as 30 features. For instance, in one cluster located in the left posterior temporal gyrus, pruning improved the Spearman correlation from 0.03 to 0.19 in out of sample data, with an average of only 28 features selected per fold. There was no case where pruning under-performed the complete, full feature set. In most cases, the improvement was considerable, though there were a few cases with moderate improvements (e.g. from -0.05 to only 0.05 in an activation cluster located around the left anterior cingulate contex, ACC).

Averaging over all 22 clusters, pruning the GloVe embeddings resulted in an increase in the mean Spearman correlation from $M = 0.025 \pm 0.056$ to $M = 0.314 \pm 0.083$, using an average of $46.95 \pm 19.6$ features. When applying leave-one-out cross-validation (LOOCV), pruning similarly enhanced the prediction for held-out data, increasing the mean correlation from $M = 0.031 \pm 0.044$ to $M = 0.143 \pm 0.069$, with an average of $49.17 \pm 17.17$ features selected per fold.

Table 1: Summary of Brain Area and pruning results when applied to entire datasets (Complete dataset) or in LOOCV context (Cross Validation), 'f' = features.

| Brain Area | Complete Dataset | | | Cross Validation | | |
|---|---|---|---|---|---|---|
| | All f | Pruned f | Features # | All f | Pruned f | Features # |
| R. Occip. Par. | 0.11 | 0.36 | 32 | 0.11 | 0.21 | 69 |
| R. Post. Par. | 0.03 | 0.37 | 42 | 0.01 | 0.18 | 44.8 |
| R. Temp. Par. | -0.04 | 0.26 | 24 | -0.01 | 0.11 | 32.8 |
| L. Sup. Occip. | 0.03 | 0.42 | 53 | 0.05 | 0.20 | 56.5 |
| L. Inf. Med. Front. (ACC) | -0.10 | 0.18 | 45 | -0.05 | 0.05 | 57 |
| R. Fusiform Temp. Occip. | 0.15 | 0.43 | 76 | 0.12 | 0.24 | 77.6 |
| R. Temp. Par. | 0.02 | 0.37 | 28 | 0.08 | 0.21 | 27.6 |
| R. Inf. Postcentral | 0.11 | 0.33 | 73 | 0.07 | 0.17 | 74.4 |
| R. Occip. Par. Sulc. | -0.05 | 0.20 | 46 | -0.03 | 0.01 | 32 |
| R. Inf. Temp./Occip./Cerebel. | -0.04 | 0.30 | 17 | -0.04 | 0.06 | 21.6 |
| L. Postcentral G. | 0.03 | 0.27 | 76 | 0.03 | 0.18 | 73.25 |
| L. Parietal, SMG | 0.01 | 0.30 | 41 | 0.03 | 0.10 | 38.9 |
| R. Mid. Temp. | 0.05 | 0.43 | 78 | 0.06 | 0.22 | 74.55 |
| L. Parieto-Occip. | 0.02 | 0.40 | 68 | 0.02 | 0.17 | 48.4 |
| R. Mid. STG | 0.01 | 0.27 | 38 | 0.02 | 0.04 | 26.4 |
| R. Post. STS | 0.09 | 0.40 | 48 | 0.08 | 0.24 | 50.9 |
| R. IFG | -0.01 | 0.22 | 39 | -0.02 | 0.07 | 43.85 |
| L. Post. STG | 0.02 | 0.37 | 21 | 0.03 | 0.19 | 28.1 |
| L. Post. STS | 0.05 | 0.31 | 78 | 0.05 | 0.11 | 64.7 |
| R. Post. Parietal | 0.02 | 0.23 | 50 | 0.03 | 0.09 | 51.6 |
| R. Mid. Temp. Sulc. | 0.0 | 0.14 | 20 | 0.0 | 0.10 | 51 |
| R. ITG | 0.04 | 0.36 | 40 | 0.04 | 0.20 | 36.86 |

From the perspective of learning neurobiological representations, our results constitute a significant advance, as the modal current approach to studying semantic spaces with word embeddings is to use the entire set of features (Complete Dataset results, all features, in Table 1). Indeed, the values we report for the non-pruned embeddings, with a mean of around 0.025 and a maximum of around 0.15 are typical of alignment values computed between DNN and brain similarity matrices in prior studies (e.g., Fernandino et al., 2022). Pruning improved the correlation value significantly, in several cases identifying meaningful correlations ($\rho > 0.34$) even when the full embeddings identified little or no correlations. This means that without pruning, one would conclude that the brain region in question cannot be predicted by GloVe representations, whereas in fact, prediction is completely possible if the correct subspace is identified, a point we return to in the Discussion.

Each pruning solution produces a subset of features specifically learned for each activation cluster, denoted as $\mathbf{F}_c$, which contains the indices of the GloVe features selected by that activation cluster. By examining all 22 $\mathbf{F}_c$ sets, we examined if there was a consistent subset of features that were either retained or excluded across the pruning solutions. As shown in Appendix Figure 5, many features were consistently excluded. Of GloVe's 300 features, 50 were never selected in any of the 22 solutions, while another 50 were selected only once. There was also weak evidence for consistent inclusion of features across multiple pruning results. No feature appeared in more than 17 of the 22 solutions, indicating the absence of a core set of GloVe features that was consistently selected across all activation clusters.

To understand the semantics of the excluded features, we analyzed the GloVe embeddings for the 534 words collected by Binder et al. (2016), and identified the top 50 words with the highest summed activation scores for these feature indices. Among these, 49 were nouns, only one was an adjective, and none were verbs. Of the 49 nouns, 10 were related to human concepts, including school, family, college, grief, moral, voter, gasp, snub, priest, and grievance. The remaining nouns were primarily animals and artifacts. The absence of adjectives and verbs from this high-scoring list suggests that the excluded features weakly emphasize human actions or activities. For comparison, we identified a set of features that tended to occur relatively frequently across pruning solutions (in 10 or more solutions), The top 50 words associated with these features were more closely associated with human activities. Verbs (18) and adjectives (5) were more prominent. (e.g., played, listened, helped, aggres-

sive, friendly). For the 27 nouns, 14 were human-related, including symphony, cathedral, banker, church, and hospital. These results already that those GloVe features most relevant for representing brain activity for the words used by Mitchell et al. (2008) are those associated with dynamic, human-related activities or attributes, rather than static concepts or objects.

## 3.2 PROBING RESULTS

The results of out of sample prediction of human annotations from GloVe embeddings using PLSR, when using the complete set of 300 features are presented in Figure 4. These serve as reference values for subsequent analyses. As detailed in the Methods, for each of the 22 prototypical activation clusters, we used only the GloVe features selected by pruning for optimizing prediction of that cluster's similarity matrix. That is, we used the feature subsets learned from pruning (indices of features appearing in column 'Complete Dataset, Feature 1' in Table 1). The results of these 22 separate analyses are presented in Figure 2, where the clusters are presented sorted in order of average prediction efficacy.

We first observe that different brain activation clusters selected for features with different levels of relevant information. Some clusters presented very little predictive capacity, whereas others approached that seen for the full feature set (normalized values approaching 0.9). As can be seen, some GloVe subsets contained information sufficient for predicting sensory features (particularly visual) but not higher level social and emotional features. Some clusters code for Vision more precisely than Audition, and some present the opposite pattern. Quite a few feature-subsets contained information sufficient for predicting Cognition, Communication, Human and Social dimensions, and these clusters also contained information about sensory dimensions. There appears to be trend where coding of Somatic and Audition features is found in clusters that also track Vision information (but not vice versa). In general, Spatial and Temporal dimensions appeared to be relatively less-well predicted. These finding show that while pruning often identified a fraction of the total GloVe features, these were sufficient to effectively predict human judgments, especially for sensory features (though some regions also appear to code for social/cognitive aspects).

As mentioned, we also identified 50 features that were consistently pruned and therefore not part of any pruning solution. To evaluate their semantics, we used this set in the probing analysis by limiting the GloVe embeddings used to these features alone. This too produced a 65-valued result indicating the correlation between the predicted ratings and ground-truth ratings (across 534 words) for each of the 65 semantic dimensions. The predictions afforded by these 50 features were, on average quite poor, particularly for the social and cognitive semantic dimensions. Interestingly, these consistently pruned features produced the most accurate prediction of the *Needs* dimension, which coded for "someone or something that would be hard for you to live without" . This dimension is important for distinguishing between small artifacts, but relatively unrelated to the nouns used by Mitchell et al. (2008). In all, this suggests that GloVe features that are not relevant to predicting the brains representational spaces (across multiple, independent activation clusters) contain relatively impoverished psychologically relevant information.

## 3.3 GRAPH ANALYSIS OF FEATURES RETAINED BY PRUNING (AIM 3)

### 3.3.1 SIMILARITY TASKS

We constructed a graph from the features retained by the 22 applications of supervised pruning. The best partition of the graph produced 3 communities, with 90, 85, and 74 features respectively. For each community, we looked at its performance, when used alone, on a similarity-prediction task.

An analysis of Wordsim-353 (Agirre et al., 2009), suggests that Community1 contained highly relevant semantic information, whereas Community3 was least relevant. Specifically, baseline prediction (using the full feature set) was $\rho = 0.658$, and similarity prediction from the three communities when used alone was $\rho = 0.652, 0.58, 0.56$. Thus, the subset of features in Community1 closely matched that of the full feature set, and also provided a substantial improvement over the prediction provided by Community2, even though the latter contained only five less features.

To complement this analysis, we conducted an ablation study, removing each community from the full GloVe feature set and measuring the resulting prediction. This ablation results were, $rho = 0.62, 0.657, 0.655$ respectively. Here, removal of Communities 2 and 3 produced performance that

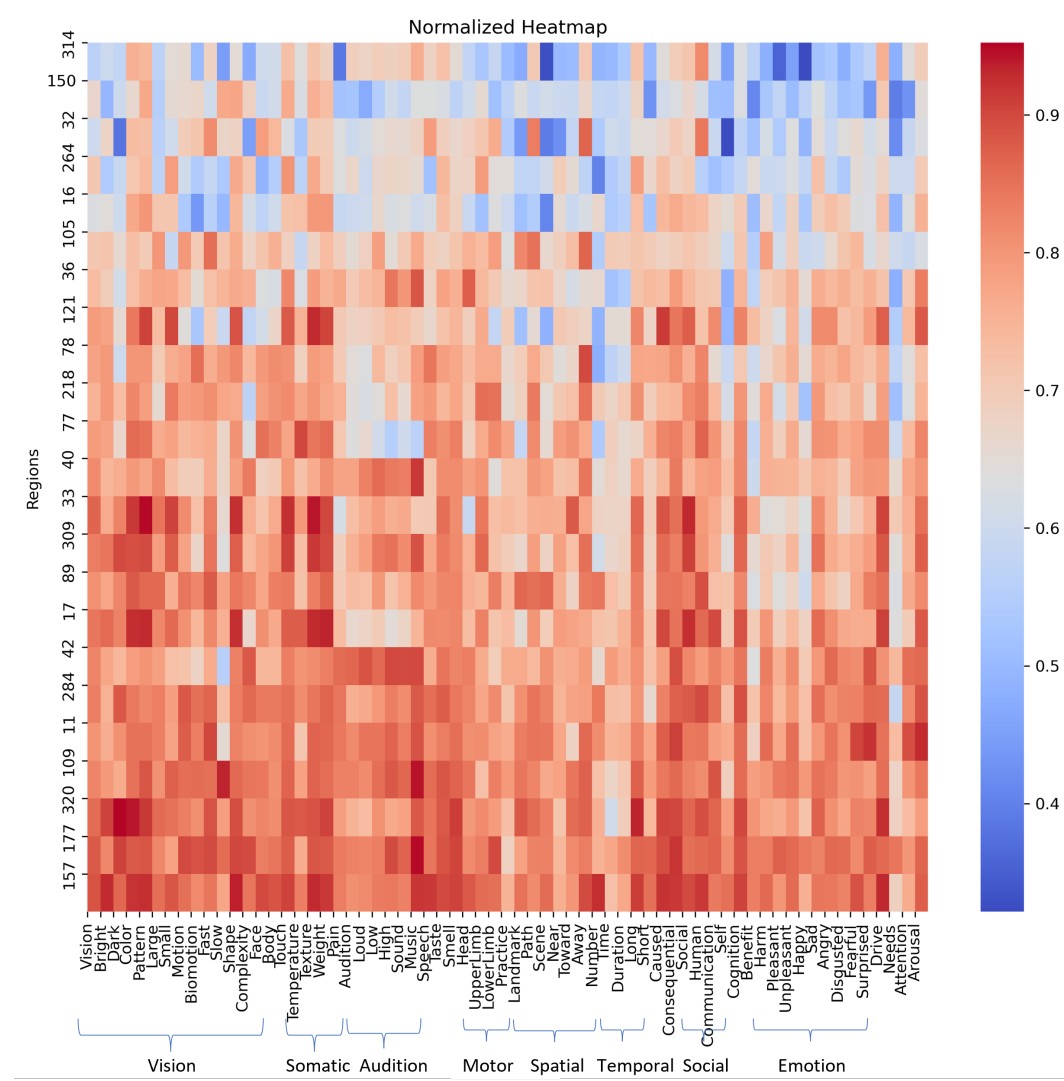

Figure 2: Prediction of each of Binder's 65 features from GloVe features optimized to predict proto-typical activation clusters across the brain. Each row on the vertical axis represents a brain cluster, and its ability to predict each of the 65 human-annotated features, through supervised pruning, is indicated on the horizontal axis.

matched baseline levels. Taken together, the data suggest that Community1 appeared to contain more relevant information, whereas Community3 appeared to contain less relevant semantic information, as its predictive power was low.

For Simlex-999 (Hill et al., 2015) the baseline was lower than Wordsim-353, $\rho = 0.407$. None of the communities surpassed baseline when used alone, though Community2 approached it ($rho = 0.35, 0.39, 0.39$ respectively). Ablation indicated that in all cases, when a community was removed, the remaining features matched or slightly surpassed baseline performance ($\rho = 0.412, 0.395, 0.392$ respectively). Thus, no clear conclusions can be made for this dataset.

### 3.3.2 ANALOGY TASKS

Figure 3 shows the results for the five semantic and eight syntactic analogy tasks, normalized to the performance using the full-feature performance. In no case did the features from any single community outperform the full feature set. However, there were some important differences across

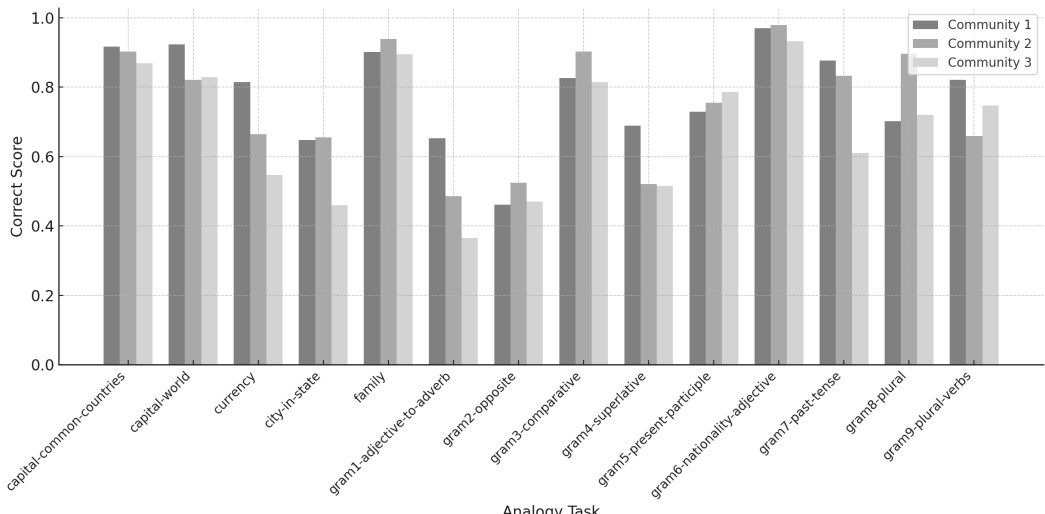

Figure 3: Performance on 14 analogy tasks for three communities produced from features retained via pruning. 'Correct score' values indicate percentage of correct response normalized by performance when using the full-feature scores.

tasks: for some analogies, some communities approached full-set performance (e.g., Community 1 for country-adjective, "Italy : Italian :: France : ?").

However, for other tasks (e.g., opposites, adjective-to-adverb) the performance was much weaker than the full-set performance. This suggests that some analogy tasks may be encoded by relatively limited sets of features that capture structured relational information. In contrast, representing opposition (antonymy) likely requires much more distributed information because they reflect relatively subtle semantic distinctions that can be spread across many different knowledge domains. As also seen in Figure 3, with the exception of one task, Communities 1 and 2 consistently outperformed Community 3, which may be expected given they contained more features. Community 3 however performed best in generating present participles (adding '-ing'). There was no clear pattern in the relative performance of features in Communities 1 and 2, though in some cases they produced quite different performance.

We also evaluated the impact of removing each community on analogy tasks. For each task, we first evaluated if the removal produced weaker performance than inclusion, but no such instances were found. We then examined if removal of a community produced better performance than baseline. There were a few such cases, though the overall effiects were minor. For Community 1, the grammar-plural task was performed better when removed (Acc = 0.79 vs. 0.77). For Community 2, the grammar-plural-verbs task was performed better when removed (Acc = 0.61 vs. 0.60). For Community 3, the currency, nationality-adjective and past-tense tasks were performed better when removed (respectively; Acc = 0.16 vs. 0.15; 0.926 vs 0.925; 0.64 vs. 0.62). The results suggest that Communities 1 and 2 contain information unrelated to grammatical structure, as their removal produced above baseline performance on such tasks.

## 4 DISCUSSION

We introduced a novel, effective, and conceptually simple approach to modeling and interpreting neurobiological representations. Using pruning, we learn a black-box encoder that aligns with the brain's representational space, and through probing, we interpret the semantic content embedded in the encoder. Our results demonstrate the effectiveness of this approach: pruning significantly improves the ability to model brain representations while probing allows to interpret this space. We also find that different brain regions have markedly different representations. Although we focused on a single language model in this study, the method is easily extended to combine features across multiple language models. These extensions are a viable direction for future work.

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

# A APPENDIX

## A.1 INFORMATION CODED IN BINDER ET AL.'S FEATURES

| Domain | Component | Description |
|---|---|---|
| Vision | Vision | something that you can easily see |
| Vision | Bright | visually light or bright |
| Vision | Dark | visually dark |
| Vision | Colour | having a characteristic or defining colour |
| Vision | Pattern | having a characteristic or defining visual texture or surface pattern |
| Vision | Large | large in size |
| Vision | Small | small in size |
| Vision | Motion | showing a lot of visually observable movement |
| Vision | Biomotion | showing movement like that of a living thing |
| Vision | Fast | showing visible movement that is fast |
| Vision | Slow | showing visible movement that is slow |
| Vision | Shape | having a characteristic or defining visual shape or form |
| Vision | Complexity | visually complex |
| Vision | Face | having a human or human-like face |
| Vision | Body | having human or human-like body parts |
| Somatic | Touch | something that you could easily recognize by touch |
| Somatic | Temperature | hot or cold to the touch |
| Somatic | Texture | having a smooth or rough texture to the touch |
| Somatic | Weight | light or heavy in weight |
| Somatic | Pain | associated with pain or physical discomfort |
| Audition | Audition | something that you can easily hear |
| Audition | Loud | making a loud sound |
| Audition | Low | having a low-pitched sound |
| Audition | High | having a high-pitched sound |
| Audition | Sound | having a characteristic or recognizable sound or sounds |
| Audition | Music | making a musical sound |
| Audition | Speech | someone or something that talks |
| Gustation | Taste | having a characteristic or defining taste |
| Olfaction | Smell | having a characteristic or defining smell or smells |
| Motor | Head | associated with actions using the face, mouth, or tongue |
| Motor | Upper limb | associated with actions using the arm, hand, or fingers |
| Motor | Lower limb | associated with actions using the leg or foot |
| Motor | Practice | a physical object YOU have personal experience using |

Table 2: Sensory and motor components, organized by domain (reproduced from Binder et al.'s Table 1)

| Domain | Component | Description |
|---|---|---|
| Spatial | Landmark | having a fixed location, as on a map |
| Spatial | Path | showing changes in location along a particular direction or path |
| Spatial | Scene | bringing to mind a particular setting or physical location |
| Spatial | Near | often physically near to you (within easy reach) in everyday life |
| Spatial | Toward | associated with movement toward or into you |
| Spatial | Away | associated with movement away from or out of you |
| Spatial | Number | associated with a specific number or amount |
| Temporal | Time | an event or occurrence that occurs at a typical or predictable time |
| Temporal | Duration | an event that has a predictable duration, whether short or long |
| Temporal | Long | an event that lasts for a long period of time |
| Temporal | Short | an event that lasts for a short period of time |
| Causal | Caused | caused by some clear preceding event, action, or situation |
| Causal | Consequential | likely to have consequences (cause other things to happen) |
| Social | Social | an activity or event that involves an interaction between people |
| Social | Human | having human or human-like intentions, plans, or goals |
| Social | Communication | a thing or action that people use to communicate |
| Social | Self | related to your own view of yourself, a part of YOUR self-image |
| Cognition | Cognition | a form of mental activity or a function of the mind |
| Emotion | Benefit | someone or something that could help or benefit you or others |
| Emotion | Harm | someone or something that could cause harm to you or others |
| Emotion | Pleasant | someone or something that you find pleasant |
| Emotion | Unpleasant | someone or something that you find unpleasant |
| Emotion | Happy | someone or something that makes you feel happy |
| Emotion | Sad | someone or something that makes you feel sad |
| Emotion | Angry | someone or something that makes you feel angry |
| Emotion | Disgusted | someone or something that makes you feel disgusted |
| Emotion | Fearful | someone or something that makes you feel afraid |
| Emotion | Surprised | someone or something that makes you feel surprised |
| Drive | Drive | someone or something that motivates you to do something |
| Drive | Needs | someone or something that would be hard for you to live without |
| Attention | Attention | someone or something that grabs your attention |
| Attention | Arousal | someone or something that makes you feel alert, activated, excited, or keyed up in either a positive or negative way |

Table 3: Spatial, temporal, causal, social, emotion, drive, and attention components (reproduced from Binder et al.'s Table 2)

## A.2 Pruning algorithm

---

**Algorithm 1** Pruning

---

1: **Inputs:**
2: $SM_{HM}$: Similarity Matrix of human similarity judgments
3: $SM_{DNN}$: Similarity Matrix of similarity estimations derived from the DNN by computing the cosine similarity between the embeddings of two words
4: **Step 1: Compute baseline**
5: Compute baseline Spearman's rank correlation $\rho(SM_{HM}, SM_{DNN})$, from the full set of features
6: **Step 2: Rank features**
7: Substep 1: Remove the first feature from all the original embeddings and compute the corresponding similarity matrix $SM_{DNNRED}$
8: Substep 2: Compute the difference $D = \rho(SM_{HM}, SM_{DNN}) - \rho(SM_{HM}, SM_{DNNRED})$. $\rho$ is Spearman's rank correlation. Higher positive values for $D$ indicate that the removed feature was important
9: Substep 3: Repeat the step above for all the possible $N-1$ feature subsets (where $N = 4096$)

10: Substep 4: Rank the features based on $D$
11: **Step 3: Construct pruned embeddings**
12: Substep 1: Starting from an empty set of features, construct pruned embeddings by iteratively reinserting one feature at a time, in descending order of importance
13: Substep 2: Compute Spearman $\rho$ after each feature reinsertion and store the values in the array $a$
14: Substep 3: Compute the maximum of $a$. Its position (index) within the array delimits the set of features to be included in the pruned embeddings

---

## A.3 SUPPLEMENTARY FIGURES

Figure 4: Prediction of each of Binder's 65 features (534 values per feature) from GloVe's 300 embedding dimensions when using leave-one-out cross-validation (LOOCV). Correlations are the Spearman rho ($\rho$) values.

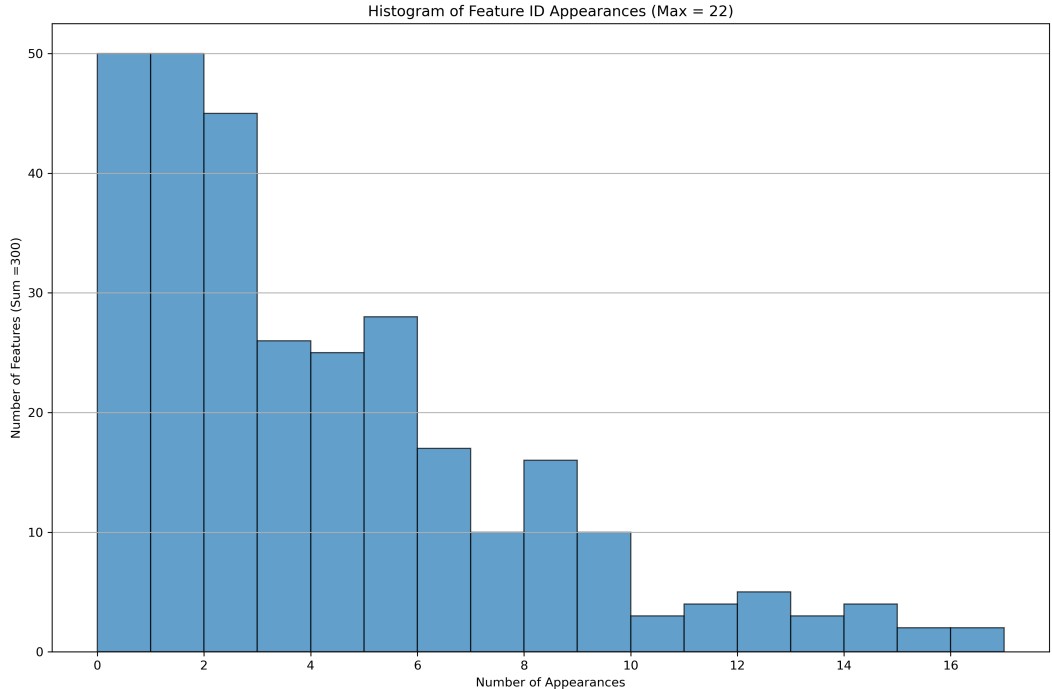

Figure 5: Histogram of feature inclusion in pruned solutions. Fifty features did not appear in any of the 22 pruning solutions, 50 appeared in only one of the solutions, and none appeared in more than 17 of the 22 solutions.

