# OpenReview forum: "Learning and Interpreting Multiple Representations of Semantics in a Neurobiological System"
_ICLR.cc/2025/Conference — ICLR 2025 Conference Withdrawn Submission_

### Official Review · Reviewer_5RGE · 2024-10-24

**Soundness:** 1
**Presentation:** 2
**Contribution:** 1
**Rating:** 1
**Confidence:** 4

**Summary:**

The authors present a novel approach for modeling the semantics of brain regions. In their approach, they first prune features from a GLoVe-derived embedding matrix and evaluate the similarity between the pruned-GLoVe-feature matrix and 22 different regions in the human brain. They then interpret the semantics of each brain region from a set of human-annotated semantic feature values.

**Strengths:**

The method introduced is well reasoned and does seem like it may be useful for generating insights into the semantics of brain regions.

**Weaknesses:**

The authors present two claimed benefits of their approach: interpreting semantic representations of a Black Box LM and enabling semantic interpretation of brain regions. While these are worthy goals and the approach is innovative, my excitement about their approach is limited by the results shown. Regarding model interpretability, the authors state that it aids in interpreting the semantics of LMs. I assuming what they mean is that it aids in our understanding of the semantic representations that align LMs and the brain. Is that a correct assumption? In any case, but especially if I have misunderstood, more clarity is needed here.


Regarding the neuroscientific implications, while this method may show promise for understanding the brain, the current manuscript would benefit from additional context and analyses to demonstrate that it does yields novel insights into the brain rather than confirming existing findings through a new methodology. For instance, it is well established that there are functional specialization of regions of visual, auditory, and social content, which are some of the clusters shown in Figure 2. While this method uncovers these known broad organizing principles of the brain, the method doesn't even seem to uniquely map to these clusters to unique regions of cortex (e.g., most regions heavily predict visual semantic representations). To me, this may suggest that semantic representations are broadly represented in cortex, which is consistent with some prior work (Huth et al., 2012; Huth et al., 2016), but the authors do not make this claim. The authors have minimally engaged with an extensive scientific literature on language and semantic representations in the brain, and it is not possible to evaluate the contributions of the method without establishing that it (1) is consistent with known semantic representations in the brain (e.g., primary auditory cortex should respond to hearing a word but not semantics of the word, similar for EVC if reading) and (2) reveals additional insights into semantic representations of the brain. It is possible that the current manuscript has accomplished these two requirements to demonstrate that the method is useful, but, in it's current form -- in the absence of contextualization within the cognitive science literature -- it is not possible to evaluate the contributions of the method.

Huth, A. G., de Heer, W. A., Griffiths, T. L., Theunissen, F. E., & Gallant, J. L. (2016). Natural speech reveals the semantic maps that tile human cerebral cortex. Nature, 532(7600), Article 7600. https://doi.org/10.1038/nature17637

Huth, A. G., Nishimoto, S., Vu, A. T., & Gallant, J. L. (2012). A Continuous Semantic Space Describes the Representation of Thousands of Object and Action Categories across the Human Brain. Neuron, 76(6), 1210–1224. https://doi.org/10.1016/j.neuron.2012.10.014


Additionally, some sections of the methods and results could be edited to improve readability, though this is a minor concern that does not affect the paper's core contributions.

**Questions:**

The following questions/suggestions are ones that the authors may wish to address in a future version of their paper but do not impact my overall evaluation of the paper.

Why does S_c use a different similarity metric than S_GLoVe?

This sentence on Line 244 on page 5 “For each of the 22 clusters we implemented pruning in ways stages” is not grammatical. Did you just mean “For each of the 22 clusters we implemented pruning in stages”?

What is being shown in Table 1? The table legend says that f = features, but I don’t follow. I gather from the text that it is the Spearman correlation, but this needs to be better specified.

All of the brain areas should be spelled out before being abbreviated.

How were the fMRI preprocessed? Did the authors fit a GLM to estimate the response for individual words or was some other method used to account for the hemodynamic delay? More details are needed here.

Did participants in the original fMRI study read or hear the words?

---

### Official Review · Reviewer_aWkX · 2024-11-02

**Soundness:** 2
**Presentation:** 1
**Contribution:** 1
**Rating:** 3
**Confidence:** 4

**Summary:**

The paper addresses an interesting topic -- functional specialization for semantic concepts and their topographic organization in the brain. It uses a well-studied albeit relatively old dataset from Mitchell et. al, 2009, to study the brain's representation of 60 nouns. The authors first fit voxel-level encoding models to predict neural activity across the whole brain. They then cluster the well-predicted voxels spatially. Then, they look to group these clusters into a smaller set of representative clusters. With these 22 clusters in hand, they then study the representational similarity of each cluster with the glove word embedding model, using either the full model or a pruned model. They find that supervised pruning of glove features leads to greater representational similarity in most cases, generalizing to heldout data. Last, they attempt to interpret the differences in features selected for different clusters, to advance an understanding of topographic functional specialization across the human brain.

**Strengths:**

- The topic is of interest.

- The idea of looking at specialization by grouping nearby voxels and then looking for distinctive feature prediction across clusters is interesting and appears to hold some promise.

- It uses multiple datasets to attempt to drive some understanding.

**Weaknesses:**

- The pruning method does not appear to offer much beyond the method of feature reweighted representational similarity analysis, which is quite popular (see Kaniuth and Hebert, 2022; NeuroImage). In fact, it is essentially a particular limited case of FR-RSA, where the weights of features are either 0 or 1. The authors do not appear well aware of the literature, as only 20 references are made.

- I found the technique of using multiple different feature spaces (the 25 feature space of Mitchell et. al to fit voxel encoding models, then the full/pruned glove model to analyze similarities within clusters) to be convoluted and potentially circular.

- As the technique is not particularly novel, it is important that the authors deliver some clear novel findings about brain function. The abstract only lists one "From a neurobiological perspective, we find that brain regions encoding social and cognitive aspects
of lexical items consistently also represent their sensory-motor features, though the reverse does not hold." I did not find the case for this finding to be particularly strong. I welcome the authors to make the case more strongly.

- The figures are poorly made, certainly well below the bar of ICLR, and do not communicate much if anything that will affect how researcher's think about semantic organization in the brain. For example, while an interesting approach of clustering brain regions is used, these regions are never visualized. In the one plot that attempts to explain some differences across brain regions, the authors use arbitrary number indices for brain regions; at minimum, anatomical labels are needed. However, in 2024, it is expected that a strong paper on this topic can make elegant visualizations of the cortical surface. Practitioners in this field understand the importance of such visualizations for relating findings to pre-existing conceptual notions of cortical organization, and for driving further intuition that will affect future research.

- The authors waste precious space presenting fits to training data (see Table 1, "complete dataset", which reports the representational similarity after selecting the features that optimize to improve that representational similarity). Only the cross-validated results are worth presenting.

- Focusing on which clusters are "best" rather than what the differences in representation are between them, seems an odd choice given the motivation of the paper.

- Averaging voxels across subjects is likely to drastically reduce the granularity of the possible findings, since there is no expectation in voxel-level alignment of fine-grained conceptual information, but only larger-scale information. I believe it would be better to construct the clusters using all subject's individual data in a group-aligned space, where the same methods can otherwise be used, but individual voxel's are kept independent and not averaged across subjects.

**Questions:**

- Why is the pruning method preferable to feature-reweighted RSA?

- Why were both models needed? Why couldn't the first step of voxel encoding models be done with the glove model?

- How does this work advance our understanding of brain function?

---

### Official Review · Reviewer_pXC5 · 2024-11-08

**Soundness:** 3
**Presentation:** 2
**Contribution:** 2
**Rating:** 3
**Confidence:** 4

**Summary:**

The authors analyze an fMRI dataset of subjects viewing noun stimuli (Mitchell 2008), focusing on topographically clustered groups of voxels that were relatively well-predicted (top 50% of R2 scores) by a linear encoding model fit to GloVe embeddings. This yields a set of 22 clusters whose representational structure is subject to further analysis and interpretation. First, the authors perform "pruning" (feature sub-sampling) within the GloVe feature space to identify subsets of features that enhance the explained variance for each brain cluster over and above the baseline RSA score computed from the entire GloVe feature space. This yields modest but consistent improvement in RSA correlations, though overall explained variance is fairly low for all analyses (maximum cross-validated Spearman rho = 0.24, with several clusters' cross-validated scores falling below 0.1). This procedure yields a subset of GloVe dimensions that are associated with each of the clusters, which are then interpreted using a probing procedure. Specifically, the authors compare the similarity of feature sets across clusters, and, assess how well each of the feature sets can predict human annotations for new words. This provides a window into whether different subgroups of latent dimensions are associated with the representations of different brain areas that are responsive to noun stimuli. The authors claim that areas encoding social and cognitive aspects of lexical items consistently also represent their sensory-motor features, while the reverse does not appear to hold.

**Strengths:**

- The paper is certainly well motivated - understanding the feature dimensions that underlie brain predictivity is a core open challenge in cognitive neuroscience.
- I commend the authors for pursuing an approach that is fundamentally data-driven, avoiding prespecified hypotheses about the involvement of different brain regions, and, which does not depend on hand-crafted feature spaces.
- The writing is fairly clear overall, and the authors did an especially good job with notational choices that made it easy to track the different feature spaces and similarity matrices (etc) that are involved in their analysis pipeline. The first few paragraphs of the introduction were particularly well-written and provided a nice review of some of the challenges that are critical to keep in mind when studying brain representations.
- Focusing on GloVe embeddings has a particular benefit, given that many recent approaches seek to explain brain data using much larger/deeper LLM models, without first assessing how much variance can be explained by simpler word embedding models.
- The sequence of the 3 main analysis aims is sensible and in principle this sort of pipeline (relying on pruning or ablating encoding models) could indeed support new forms of insight, especially when applied to richer datasets with more samples of diverse stimuli, and to encoding models that achieve higher overall predictivity than those presented here.

**Weaknesses:**

With the caveat that language is not my primary subfield within cognitive neuroscience, I can say that several core weaknesses leave me unable to rate the paper higher than a 3, currently. Despite the solid motivation and methodological soundness, my read is that the models/datasets and corresponding results do not seem rich enough (in terms of explanatory power) to support new insight into the nature of semantic representations in the brain, let alone the deep and challenging question of how representational dimensions in language-responsive cortex (the encoding of information) relate to functions and behaviors such as reading, writing, and speaking.

- The Mitchell 2008 dataset is a fair starting point for the authors' analyses, but the paper would have been much stronger if more recent language fMRI datasets were taken into account, such as:
Pereira et al. (2018), Nastase et al. (2021), Tuckute et al (2024)... any of these would have offered a more diverse array of stimulus content (and perhaps better SNR...) to support more compelling LM prediction levels. A fundamental issue with the paper is that prediction levels are so low (as mentioned above), and there is no report of noise ceilings to help the reader understand whether even low scores should be considered meaningful.  Relatedly, on the modeling side, I wonder if models other than GloVe could confer better prediction scores (and consequently, more useful interpretations) of the brain dimensions. In principle, the authors' analysis framework would be applicable in a fairly straightforward way to more performant computational models of language.

- Glaring lack of relevant lit review: *no* papers from Ev Fedorenko, Greta Tuckute, Mariya Toneva, Leila Wehbe, nor Alex Huth were cited or discussed in this work. These individuals, among others, have pioneered the effort in recent years to understand dimensions of language representation across different groups of language-responsive brain voxels. Much more thorough citation of work from these groups would help situate this paper better and contextualize its findings, especially for readers who may not be experts in language cog neuro.  Other highly relevant work that explores the use of GloVe embeddings to predict neural data is [this CCN paper](https://www.researchgate.net/profile/Colin-Conwell/publication/373731750_The_Unreasonable_Effectiveness_of_Word_Models_in_Predicting_High-Level_Visual_Cortex_Responses_to_Natural_Images/links/65fcfe1ad3a08551423d2db3/The-Unreasonable-Effectiveness-of-Word-Models-in-Predicting-High-Level-Visual-Cortex-Responses-to-Natural-Images.pdf).  Given that several occipital cortical areas show up in Table 1, [this line of work](https://arxiv.org/abs/2209.11737) is also highly relevant.

- An issue with framing: purely representational analyses like those the authors pursue here are not sufficient to "identify functional brain modules", under the assumption that function refers to some behaviorally-relevant output. This point is easily illustrated in the classic signature observed in visual fMRI, where [chairs and other objects can be linearly decoded from the fusiform face area](https://zora.uzh.ch/id/eprint/3224/9/haxby_science2001V.pdf), whose functional role is causally linked much more to face processing than chair processing. The representational geometry of a region alone is not sufficient to understand how the decodable information there is put to use functionally. Different paradigms (esp those that involve causal perturbation) are required to make functional claims.

- Unless I am mistaken, the pruning/subsampling approach appears to be a special case of the "voxel-encoding RSA" ([veRSA](https://www.nature.com/articles/s41467-022-28091-4) technique), where subsets of features from a model are combined to predict the activity pattern of each voxel. Then, using held-out test stimuli, predicted voxel activations are generated, transformed into RDMs, and correlated with empirically measured RDMs from the same stimuli. In the present work, instead of using continuous regression weights to mix the GloVe features together, a binary feature subselection is applied to optimize RSA scores, effectively forming a binary mask that retains or excludes specific features. This binary mask acts as a sparse linear transformation on the model features, encoding only those most aligned with the brain's representational structure. By iteratively selecting subsets of features that maximize RSA alignment, this subselection approach indirectly mirrors voxel encoding, as it performs an encoding-like operation on representational similarity rather than directly predicting voxel activity. This veRSA procedure is known to [dramatically increase RSA scores](https://www.nature.com/articles/s41467-024-53147-y) above vanilla/classical (unweighted) RSA, and the fact that predictions improve in Table 1 therefore seems like more of a sanity check than a "significant advance" in our understanding of neurobiological representations (lines 351-352).

- For this kind of paper, it's usually helpful to provide some visualizations of the brain to show areas that are implicated in different analyses/findings. I would consider doing so in future revisions to help the reader understand where the key clusters lie across cortex. Additionally, there is no discussion of how the "Brain Areas" in Table 1 are derived from the analysis procedure.  This detail should be added, since as far as I can tell there was no constraint indicating that cluster boundaries should conform to pre-defined ROIs. Table 1 is also missing that the key outcome metric is Spearman rho, rather than R2.

- To emphasize again: it is critical that the authors contextualize their correlation values with appropriate noise ceilings so that the reader can assess any further claims regarding the dimensions of the clusters, and how they may form network communities. Without noise ceilings, it is impossible to critically assess many of the claims toward the end of the paper.

- The conclusion is far too brief - I would recommend streamlining the methods detail (punting some to the Appendix) in order to leave room for more summary, discussion of the results and their implications, a section on the limitations of this work, etc...

**Questions:**

To summarize some of the key points above:

- What is the SNR of the Mitchell data, and what are the noise ceilings on the RDMs from each cluster? What is the R2 cutoff used to identify the 50% of voxels for subsequent analysis? What are non-null data - how is this defined? Little details like this really matter in interpreting the paper.
- How do the claims here relate to or differ from the last decade-plus of research mapping computational language models to fMRI data to understand dimensions of linguistic representation? Would similar dimensions and clusters be identified using more recent publicly-available datasets, and/or, by using models that can achieve higher prediction levels of cluster RDMs relative to their noise ceilings?

In principle I would be willing to raise my score if several of the key methodological points above were addressed during the review process, and if the overall narrative were refined to leave much more room for discussion.

---

### Note · Authors · 2024-11-15

**Comment:**

We thank the reviewers for the careful reading and useful comments.

**Withdrawal Confirmation:**

I have read and agree with the venue's withdrawal policy on behalf of myself and my co-authors.